# Advances in the Application of Nanomaterials as Treatments for Bacterial Infectious Diseases

**DOI:** 10.3390/pharmaceutics13111913

**Published:** 2021-11-12

**Authors:** Yuan-Pin Hung, Yu-Fon Chen, Pei-Jane Tsai, I-Hsiu Huang, Wen-Chien Ko, Jeng-Shiung Jan

**Affiliations:** 1Department of Internal Medicine, Tainan Hospital, Ministry of Health and Welfare, Tainan 70007, Taiwan; yuebin16@yahoo.com.tw; 2Department of Internal Medicine, National Cheng Kung University Hospital, College of Medicine, National Cheng Kung University, Tainan 70403, Taiwan; 3Department of Medicine, College of Medicine, National Cheng Kung University, Tainan 70101, Taiwan; 4Master of Biomedicine Program, National Taitung University, Taitung 95092, Taiwan; yufons@gmail.com; 5Department of Chemical Engineering, National Cheng Kung University, Tainan 70101, Taiwan; 6Department of Medical Laboratory Science and Biotechnology, College of Medicine, National Cheng Kung University, Tainan 70101, Taiwan; peijtsai@mail.ncku.edu.tw; 7Center of Infectious Disease and Signaling Research, National Cheng Kung University, Tainan 70101, Taiwan; 8Department of Pathology, National Cheng Kung University Hospital, Tainan 70101, Taiwan; 9Department of Biochemistry and Microbiology, Oklahoma State University Center for Health Sciences, Tulsa, OK 74107, USA; ihsiuhuang@gmail.com

**Keywords:** nanoparticle, silver nanoparticle, biofilms, implants, bacterial infection, polymers

## Abstract

Bacteria-targeting nanomaterials have been widely used in the diagnosis and treatment of bacterial infectious diseases. These nanomaterials show great potential as antimicrobial agents due to their broad-spectrum antibacterial capacity and relatively low toxicity. Recently, nanomaterials have improved the accurate detection of pathogens, provided therapeutic strategies against nosocomial infections and facilitated the delivery of antigenic protein vaccines that induce humoral and cellular immunity. Biomaterial implants, which have traditionally been hindered by bacterial colonization, benefit from their ability to prevent bacteria from forming biofilms and spreading into adjacent tissues. Wound repair is improving in terms of both the function and prevention of bacterial infection, as we tailor nanomaterials to their needs, select encapsulation methods and materials, incorporate activation systems and add immune-activating adjuvants. Recent years have produced numerous advances in their antibacterial applications, but even further expansion in the diagnosis and treatment of infectious diseases is expected in the future.

## 1. Introduction

Antimicrobial resistance in bacteria has become a substantial threat to human health in recent years; however, few new antimicrobial agents with innovative mechanisms of action are available to effectively combat these bacteria [1]. Moreover, many bacteria form complex communities called biofilms in their environment; biofilms provide advantages to bacteria, such as enhanced resistance toward antibiotic challenge [2]. Nanomedicine is a category that means the application of nanotechnology in medical conditions and is defined as the practice of nanotechnology for the detection, prevention, and therapy of diseases, for example infectious diseases [3]. One of the most commonly used nanomaterials in nanomedicine is nanoparticles (NPs), which show highly tunable physical and optical properties and the ability to produce a wide library of compounds [3]. A variety of organic, inorganic, and hybrid NPs have been synthesized using various approaches in the past few decades. Nanomaterials comprised of different organic and/or inorganic constitutes are defined as hybrid nanomaterials. The most notable example is organic-inorganic hybrid NPs, which are commonly prepared by absorbing or grafting polymers onto inorganic NPs. For example, several studies have demonstrated that hybrid NPs based on silver (Ag) NPs are promising in the therapy of antimicrobial-resistant bacterial biofilm-related infectious diseases [4,5].

Nanomaterial-based therapies represent promising tools to fight against bacterial infections that are difficult to treat, for example, those characterized by biofilm formation or antimicrobial resistance [6]. In this review, we introduce the different types of nanomaterials and highlight their current applications as treatments for bacterial infections characterized by possible antibiotic resistance and biofilms. In this review article, we emphasize the importance of nanomaterials in combating bacterial infections in different body systems.

## 2. The Development of Organic, Inorganic and Hybrid NPs for Infectious Diseases

Bacteria-targeting nanomaterials have been widely used to treat bacterial, fungal, viral and parasitic infectious diseases in recent years [7,8,9,10,11,12,13,14], and their applications include direct bactericidal effects [15,16,17,18], encapsulation of probiotics [19], and encapsulation of bactericidal agents [20].

Inorganic nanomaterials with direct bactericidal effects include calcium silicate (CaSi) [15,21], silver (Ag) [22,23], copper (Cu) [24,25], and gold (Au) NPs [17,18]. Among the inorganic nanomaterials, Ag NPs have been one of the most widely used antimicrobial materials due to their broad-spectrum antibacterial capacity and low propensity to engender drug resistance [18]. Ag NPs had a good antimicrobial effect in both gram-positive and gram-negative bacteria, for example methicillin-resistant *Staphylococcus aureus* (MRSA), vancomycin-resistant *Enterococcus* (VRE), *Pseudomonas aeruginosa*, and *Klebsiella pneumoniae* [18,22,23]. Au NPs have been shown to decrease the effect of lipopolysaccharide (LPS), an important toxin in gram-negative bacteria that causes fulminant immune responses through its effect on dendritic cells and immune reactions [26]. Transition metal oxides and chalcogenide (TMO&C) nanomaterials have been characterized as having excellent physiochemical, electronic, and optical properties. They possess a functional architectural assembly, proving their benefits in combating bacterial infections [27]. Similarly, magnetic iron oxide NPs (MIONPs) exhibit improved adherence to *Staphylococcus* and extracellular polymeric substances, thus increasing bacterial biofilm eradication through the encapsulation of antimicrobial agents [28]. In summary, inorganic nanomaterials have been widely used to treat various bacterial infectious diseases, according to the characteristics of inorganic materials. Notably, the intrinsic toxicity of some inorganic materials (i.e., heavy metals) and their ability to accumulate and persist in the human body are of great concern when used in clinical patients [29]. Successful clinical application of these NPs primarily depends on their stability, circulation time, access and bioavailability at disease sites, and safety profile [29].

Organic nanomaterials have also been developed for the treatment of bacterial infectious diseases in the past few decades. Among them, polymeric NPs have become an important class of nanomaterials and have recently attracted intense interest. Polymeric NPs such as micelles and vesicles are formed by the self-assembly of amphiphilic copolymers in aqueous solutions through various physical interactions, such as hydrogen bonding and electrostatic or hydrophobic interactions [30,31,32]. Chemical cross-linking of polymers or in situ polymerization of monomers in nanoconfined environments also promotes the formation of NPs, which are typically named nanogels [33,34,35]. For example, antimicrobial agents have been encapsulated in cross-linked nanogels, achieving better antibacterial activity with less cytotoxicity [36]. Tumor necrosis factor (TNF)-related apoptosis-inducing ligand (TRAIL), which triggers an extrinsic apoptotic pathway via death receptors, was encapsulated in a bactericidal polypeptide-crosslinked nanogel that suppressed *K. pneumoniae* and overactive macrophages. Interestingly, nanogel and TRAIL nanogel treatments were more toxic toward LPS-activated cells than toward naïve cells (Figure 1) [36]. The highly selective character of antimicrobial agent-encapsulated cross-linked nanogels is beneficial for increasing the local concentration of the antimicrobial agent without damaging normal tissue.

An important category is hybrid nanomaterials, and many of them have been applied for treatments of infectious disease [4,5,37,38,39,40,41,42,43,44]. There are some examples of hybrid nanomaterials:(a)Ag-based hybrid NPs: Intracellular protein with gentamycin attached was analyzed to create the biocompatible Ag NPs [5]. The created nanohybrid had been shown to be effective during gram-positive, gram-negative and drug-resistant bacteria infection [5]. A hybrid antimicrobial particle with improved antimicrobial efficacy and good aqueous dissolubility relying on Ag NPs and curcumin had been made with oxidized amylose as an environmentally-friendly NP [4]. The hybrid antimicrobial agent revealed good dissolubility in aqueous solution, improved antimicrobial efficacy, excellent antioxidant effect and better cell compatibility [4]. A hybrid hydrogel of organic collagen capped on an inorganic Ag NP and melatonin had been used to promote tissue regeneration of wounds with skin defects during multi-drug resistant bacterial infection [37]. Assemblies of organic thiolated hyaluronic acid with Ag NP hydrogels were effective against *S. aureus* infection in chronic wounds [38]. The hybrid NP with coating Ag on organic fluorhydroxyapatite had been shown to enhance its antimicrobial ability when applied in skeletal system implants [39].(b)Anti-bacterial agent-based hybrid: An anti-bacterial peptide attached with biomimetic phage-platelet hybrid NP was created for extended circulation in blood and enhanced anti-microbial efficacy [40]. Poly lactic-co-glycolic acid (PLGA)-based NP distribution systems were analyzed to treat the bacterial biofilm-related infections. Azithromycin (a macrolide antibiotic)-loaded carbon quantum dots (CQDs)–PLGA hybrid NPs showed chemo-photothermally synergistic anti-biofilm effects against *P. aeruginosa* biofilms [41]. Au NP combined with β-lactam antibiotic had been shown to be effective against MRSA infection [42].(c)Theranostics hybrid nanomaterial: The hybrid of inorganic material (Mn_3_O_4_ NPs) with organic citrate (C-Mn_3_O_4_) and folic acid (FA-Mn_3_O_4_) ligands were bactericidal against *S. hominis* by causing membrane rupture [43]. The multielement physical characters of these organic–inorganic hybrid materials revealed their further application both in diagnosis and therapy (theranostics) [43].(d)Bifunctional hybrid nano-flowers: A bifunctional hybrid nano-flower composed of an organic part (the rabbit polyclonal antibody of *H. pylori*) and inorganic part (the Cu NP), prepared with an enzyme-linked immunosorbent assay, was invented for the detection of *H. pylori* infection [44].

Overall the hybrid nanomaterials showed enhanced antibacterial activity with less adverse drug reactions compared to conventional treatment.

In recent years, nanostructured polymers have also been broadly applied as treatments for infectious disease [45,46,47,48,49]. The preparation of polymeric nanomaterials with nanoscale architectures, such as star-shaped polymers, can be achieved by exploiting polymer topology. Among these materials, star-shaped polymers provide distinct advantages over their linear counterparts. Star-shaped polymers are typically synthesized using dendrimers as initiators. The star-shaped topology effectively decreases the tendency of the cationic polymers to interact with other cells, reducing cytotoxicity when the arms are “locked” in a nanoparticle form [45,46,47,48,49]. The so-called ‘structurally nanoengineered antimicrobial polymers’ (SNAPs) are different from existing self-assembled antimicrobial macromolecules, which dissociate into monomers below their critical micelle concentration [39]. Since 2016, SNAPs have shown great potential as low-cost, effective antimicrobial agents that may represent a tool for combating multiple-drug resistant (MDR) pathogens, such as MDR *Acinetobacter baumannii* [47]. Additionally, the star-shaped structure has been further modified and optimized for the subsequent development of antimicrobial agents used in clinical applications [45].

A new SNAP (14–26 nm in diameter) that contains mixtures of poly(L-lysine) (PLL) and glycopolymer arms was reported to have antimicrobial activity toward gram-positive bacteria, including MRSA and VRE, while being nonhemolytic and showing excellent mammalian cell biocompatibility [45]. Star-shaped polymers based on PLL modified with an indole group (PLL-*g*-indo) were revealed to have the best antibacterial activity against enterohaemorrhagic *E. coli* (EHEC), improving cytotoxic selectivity toward pathogens over mammalian cells without hemolytic activities (Figure 2) [48]. A hybrid SNAP synthesized by atom transfer radical polymerization (ATRP) and composed of poly[2-(dimethylamino)ethyl methacrylate] (PDMA) as the arms and polyhedral oligomeric silsesquioxane (POSS) as the core (POSS-g-PDMA) exerted antibacterial effects against *E. coli* infection in zebrafish embryos while maintaining low cytotoxicity and was used in aquaculture applications [46]. A polymer synthesized by ring-opening polymerization and composed of copolymer poly(L-lactide)-*block*-poly(phenylalanine-stat-lysine) [PLLA_31_-*b*-poly(Phe_24_-stat-Lys_36_)] chains self-assembles to form micelles in aqueous solution and has antibacterial properties against both gram-positive and gram-negative bacteria without evident antibiotic resistance while being biodegradable [49]. SNAPs are predicted to be more widely used in bacterial infections in the future to increase the efficacy and decrease adverse events compared to traditional NPs.

## 3. The Wide Application of Nanotechnology in Bacterial Infectious Diseases

### 3.1. Diagnosis/Increased Accuracy and Efficacy of Diagnostic Tests

In addition to serving as treatments, nanotechnology has improved the accuracy of diagnosing bacterial infectious diseases [44,45,46,47]. The rapid development of nanoscience and nanotechnology has been used to design nanomaterial-assisted biosensors with better detection performance for bacterial infections, as shown in the examples listed below [50,51].

#### 3.1.1. Graphene-Based Biosensors

The unique 2D structure and single-atom thickness of graphene sheets enable graphene-based biosensors to be highly flexible to any tiny changes in the surrounding environmental conditions [51]. Graphene-based biosensors have been widely used in infectious disease, especially in coronavirus (COVID-19) infection [52,53]. For example, an novel aptameric dual channel graphene-TWEEN 80 field effect transistor (DGTFET) biosensing machine with on-site analyzing of interferon (IFN)-γ, TNF-α, and interleukin (IL)-6 works as quickly as 7 min with limits of detection (LODs) and as low as 476–611 × 10^−15^ m in bio-fluids [52].

For bacterial infection, surfaces of graphene products, graphite, graphene, graphene oxide and reduced graphene oxide have been revealed to be a good platform for antibacterial effects and improved biocompatibility in tissue engineering, biosensors or antimicrobial agent-carriage [54]. A graphene (SLG)/Au NP-based biosensor has been used in the diagnosis of *S. aureus* gene sequences, with detection limits as low as 12.4 pg/mL [55]. A biosensor using reduced graphene oxide-carbon nanocomposites prepared using the hydrothermal method has been used for precise and fast label-free electrochemical detection of pathogenic bacteria, such as *S. enterica,* in a wide linear dynamic range from 10^1^ to 10^8^ cfu mL^−1^ and with a 10^1^ cfu mL^−1^ limit of detection [56]. A fluorogenic DNAzyme probe was utilized in diagnosis of *E. coli* infection and could detect with a detection limit of 1000 CFU [57]. A label-free electrical biosensor coupled with graphene exhibits accurate diagnosis of pathogenic *E.coli* O157:H7 with the sensors transduced through electric charges [58]. Thermally decreased graphene oxide-based field-effect transistor (rGO FET) coupled with an ultrathin layer of Al_2_O_3_ shows on-time diagnosis of *E. coli* bacteria [59]. The sensor has the ability to detect a single *E. coli* cell within 50 s in a 1 µL sample volume [59]. A quick detection device for *Vibrio parahaemolyticus* was developed by merging loop-mediated isothermal amplification (LAMP) and one-use electrochemical sensors with screen-printed graphene electrodes (SPGEs) [60]. The LAMP responses using primers targeting *V. parahaemolyticus toxR* gene could diagnose *V. parahaemolyticus* within 45 min with the detection limit of 0.3 CFU per 25 g of uncooked seafood [60]. A novel self-propelled micromotor-based immunoassay (MIm) has been used for C-reactive protein (CRP) detection of blood of preterm infants suspected to have sepsis [61]. Graphene sensors could be merged with various signal transduction technologies to improve their applications in the future.

#### 3.1.2. NP-Based Lateral Flow Biosensor

An assay that combines LAMP with an NP-based lateral flow biosensor (LFB), named LAMP-LFB, enables the simple, reliable, and objective detection of *Mycoplasma pneumoniae* [62]. The development of NP-enhanced antibodies and DNA sensors for detecting *Salmonella* using a microfluidic-based electrochemical system provides a rapid detection method with high sensitivity and specificity [63]. Multiple cross displacement amplification (MCDA) combined with Au NP-based lateral flow biosensors (LFBs) provides a simple, fast and sensitive method for the accurate identification of *P. aeruginosa,* among other pathogens [64]. Au/Ag hybrid NPs composed of gold nanorods (AuNRs) coated with Ag serve as excellent theranostic agents in photoacoustic (PA) imaging, as well as the treatment of bacterial infections [65]. The multiple cross displacement amplification (MCDA) and NP-based LFB (MCDA-LFB) were combined to detect *H. influenzae*, which has been proven to be reliable, rapid, and not complicated [66]. An electrochemiluminescent lateral flow immunosensor (ECL-LFI), Ru(bpy)32+-labeled AuNPs, has been used to detect CRP [67]. Another multiplex loop-mediated isothermal amplification coupled with a NP-based lateral flow biosensor (m-LAMP-LFB) has been used in detection of all *S. aureus* species [68]. The LAMP-LFB also provides a reliable and rapid assay for *E. faecalis, Neisseria meningitidis* and *M. pneumoniae* detection, which is significant for diagnosis and follow-up treatment [62,69,70].

#### 3.1.3. Sensitive Label-Free Immunosensing

An *H. pylori* immunosensor based on a platinum NPs/poly(3,4-ethylenedioxythiophene)/reduced graphene oxide (Ptnano/PEDOT/red-GOx)-modified gold electrode (Au-ET) was fabricated in a stepwise manner for the detection of cytotoxin-associated gene A antibody [71]. This novel immunosensor exhibited a good accuracy, precision and reliability, and the sensor has an excellent linear range of 0.1 ng/mL to 30 ng/mL by limiting the detection range to 0.1 ng/mL [71]. A label-free electrochemical immunosensor was developed for quick detection of *H pylori* after covalent conjugation of the antibody (CagA) on the nanomaterials altered the Au electrode [72]. An Au disc electrode coupled with anti-*P. syringae* pv. lachrymans antibodies and conjugated with bovine plasma albumin was developed and sensor development was characterized by cyclic voltammetry (CV) and antigen detection by electrochemical impedance spectroscopy (EIS) measurements [73]. A new paper-based sensing platform has been applied in a label-free potentiometric immunosensor for *Salmonella typhimurium* detection based on the blocking surface principle [74]. A photonic interferometer biosensor based on a bimodal waveguide (BiMW) was developed for the rapid and label-free detection of bacteria, such as *E.coli* directly in ascitic fluid [75].

#### 3.1.4. Nanosieving Microfluidic System

An electropolymerized self-assembled layer of Au NPs was fabricated in a portable nanosieving microfluidic system (NS-MFS). Redox-active gold NPs (raGNPs) enhanced the electrical conductivity and provided the detection limit of the device, reaching 10 CFU/mL for *P. aeruginosa* and *S. aureus* spiked in plasma [76]. A microwave-microfluidic biosensor has been developed for quick, contactless and non-invasive detection of the concentration and growth of *E. coli* [77].

#### 3.1.5. Au NPs

Au NPs have been widely used as biosensors for bacteria detection. Thiolation of the chimeric phages directed on various bacterial pathogens caused aggregation of AuNPs, resulting in a visible colorimetric reaction in front of at least about 100 cells of the target bacteria [78]. The photoelectronic Au nanostructure biosensor could detect citrullinated histone H3, an important biomarker during neutrophil cell death in bacterial infection [79]. The Au NP-based biosensor has been used to detect the colistin (an important antimicrobial agent for gram negative bacteria) resistance gene *mcr-1* [80].

#### 3.1.6. Carbon Nanotube Biosensor

This is an electrochemical biosensor that is constructed from single walled carbon nanotubes that could detect the change in intracellular hydrogen peroxide during contact with lipopolysaccharides, an important component of gram-negative bacteria [81].

These materials are some examples of using nanotechnology to diagnose bacterial infections. With the increasing application of nanotechnology in the detection of bacteria, the accuracy and efficacy of diagnostic tests in bacterial infections will increase substantially.

### 3.2. Vaccination/Vectors or Adjuvants to Deliver Antigenic Proteins

NPs can be used as vaccine vectors to deliver antigenic proteins that induce humoral and cellular immune responses [82,83]. NPs carrying outer membrane vesicles (OMVs) from carbapenem-resistant *K. pneumoniae* (CRKP) are believed to be potential vaccines for intractable CRKP infection [82]. NPs containing polysorbitol transporter (PST) and pneumococcal surface protein A (PspA) induce protective immunity against *Streptococcus pneumoniae*, with long-term reaction mediated by T helper (Th)2 or T follicular helper (Tfh) cells [83]. T and B cell immune responses are mediated by antigen-presenting cells through the activation of a peroxisome proliferator-activated receptor gamma (PPAR-c) pathway [83]. These immune reactions suggest that NPs might serve as a mucosal adjuvant in a subunit vaccine [83]. Similarly, NPs coated with membranes of extracellular vesicles secreted by *S. aureus* (i.e., NP@EV) may serve as an active-targeting antibiotic carrier [84]. NPs coated with a PEGylated lipid bilayer (i.e., NP@Lipo; PEG = poly(ethylene glycol)) are internalized by *S. aureus*-infected macrophages with higher efficiency and accumulate within major organs (such as kidneys, lungs, spleen, and heart), suggesting that they are beneficial for treating metastatic *S. aureus* infections [84]. NPs have been shown to be excellent vaccine vectors or adjuvants to deliver antigenic proteins that induce host immunity against bacterial infections.

## 4. Current Applications of Nanotechnology as Treatments for Bacterial Infections in Regard to Possible Antibiotic Resistance and Biofilms

### 4.1. Medical Device/Efficacy against Biofilms

Biofilm formation is a key step in the bacterial infection of implants, catheters and naive tissues [85]. Many bacteria, such as *S. aureus*, *E. coli* or *P. aeruginosa,* have the capacity to self-secrete a matrix of extracellular polymeric substances (EPSs) that encapsulate bacterial societies. This matrix then progresses to form an advanced 3D structure called a biofilm that functions as a protective barrier against host immunity, environmental pressure, and the inner diffusion of antimicrobial agents [85,86,87,88,89].

#### 4.1.1. Application of Nanotechnology in Biofilms in Biomaterial Implants

A common application of nanotechnology is biomaterial implants. These devices generally present a 5% failure rate, with most cases attributed to the bacterial colonization of biomaterial surfaces, formation of biofilms and spread into adjacent tissues [85,86,90,91,92,93]. *S. aureus*, for example, recognizes host extracellular matrix (ECM) molecules and initiates colonization of the epithelium through microbial surface recognizing adhesive matrix molecules (MSCRAMMs), such as fibronectin binding protein A (FnBPA) [92]. In contrast, NPs interfere with the function and binding property of FnBPA, enabling their use in a wide variety of biomedical applications [92]. Similarly, 316L stainless steel coated with a polyelectrolyte multilayer and a mesoporous bioactive glass exhibited controlled antibiotic (tetracycline) release and biocompatibility, showing great potential in endowing orthopedic implants with antibacterial and osteoconductive properties [93]. Silver/copper (Ag/Cu)-coated catheters were investigated as tools to prevent MRSA infection and biofilm formation [89].

#### 4.1.2. Improve Biofilm Penetration with Nanotechnology

Bacteria-sensitive NPs prepared from poly(lactic-coglycolic acid) and poly(Ɛ-caprolactone), which were decorated with chitosan and combined with dissolved microneedles of doxycycline, could serve as antimicrobial agents with improved biofilm penetration specifically in the delivery of doxycycline to an infection site [94]. Zeolite-based imidazole framework (ZIF-8)-covered mesoporous polydopamine (MPDA) NPs were combined with Pifithrin-μ (PES), a natural inhibitor of heat-shock protein (HSP) that is important in bacterial related destruction and inflammation [95]. The ZIF-8 shell of the MPDA@ZIF-8/PES nanoplatform could quickly degrade under the acidic microbial environment, which activated the release of PES and Zn ions [95]. In general, this penetration of biofilms to target enclosed bacteria has been an important therapeutic aim when developing NPs for bacterial infectious diseases in different organs or systems [16,85,86,87,88,89,94].

#### 4.1.3. Nanotechnology-Based Therapy in Destroying Biofilm Formation

NPs have been used as one of the therapeutic strategies for nosocomial bacterial infections [96]. A nanoscale surface coating containing silicon NPs forms a hydrophobic surface that prevented the growth of bacteria with a significant reduction in the frequency of isolation of *Acinetobacter* spp. on coated surfaces [96]. A synthetic iron NP (FeOOH-NP) was shown to be an active dose-dependent agent that inhibited the formation of *P. aeruginosa* biofilms, modulating bacterial motility and other important virulence factors related to biofilm formation [86]. Another sulfur-functionalized fullerene NP (SFF NP) prepared by chemical vapor deposition effectively inhibits biofilm formation, eradicating preformed biofilms of *P. aeruginosa* and disrupting the expression of the exotoxin A (toxA) gene [87]. Similarly, zinc oxide NPs (ZnO-NPs) have been shown to inhibit biofilm formation and flu gene expression in uropathogenic *E. coli* (UPEC) strains [88].

#### 4.1.4. Biofilm Monitor with Nanotechnology

Biofilm formation is critical in antimicrobial agent-resistance and treatment failure; however, so far there is no detection method to monitor the formation of biofilms. Anti-*S. aureus* antibodies were coupled with Au NPs to develop an immunochromatographic strip (ICS) [97]. The invented ICS could detect 10^2^ CFU/mL *S. aureus* in 15 min among patients with neonatal sepsis [97]. NP-based biosensors have been used to detect and prevent the formation of biofilms by disturbing the adhesion of bacteria to the surface of food [98]. The developed prototype ICS could detect as low as 5 µg purified polypeptide and 10^2^ CFU/mL *S. aureus* within 15 min. An optoelectronic machine by the dual array of interdigitated micro- and nanoelectrodes in parallel could detect the bacterial biofilm development by the optical and electrical monitor methods [99].

#### 4.1.5. Support of Tissue Regeneration in Biofilms

Tissue regeneration is often retarded when biofilms are formed during bacterial infections. Injectable calcium phosphate bone cement-chitosan paste containing graphene oxide has been used for endodontic therapy, with antimicrobial effect against *E. faecalis* biofilms, together with the support for the dental pulp stem cells [100]. The fast coverage of the nano-textured titanium by human mesenchymal stem cells together with stimulation of bony differentiation could decrease the chance of biofilm formation during implant integration with the human bone tissue [101]. Ag NPs coupled with polyetheretherketone have been developed for osteogenesis and angiogenesis in addition to the antibacterial ability [102]. Copper-incorporating mesoporous bioactive glass NPs have been invented for bone regeneration as well as antimicrobial ability [103].

Biofilm formation is a troublesome problem when treating bacterial infections, and NPs have been shown to be effective at inhibiting biofilm formation and eradicating preformed biofilms.

### 4.2. Application of Nanotechnology in Detecting Possible Antibiotic Resistance in Bacterial Infectious Diseases

As mentioned above the m-LAMP-LFB was applied not only to detect all *S. aureus* strains, but also to detect methicillin-resistant genes from all *S. aureus* species [68]. Multiplex MCDA (m-MCDA), which aims to detect the *nuc* gene (*S. aureus*-specific gene) and *mecA* gene (encoding penicillin-binding protein-2′), has been shown to detect *S. aureus* strains and MRSA within 85 min [68]. Fast dissemination of the colistin resistance (*mcr*) gene *mcr-1* in *Enterobacteriaceae* is very important in combating bacterial infections [80]. MCDA merged with the sensing of agents by Au NP-based LFB assay was identified to detect the mcr-1 gene with simplicity, rapidity, specificity, and sensitivity [80]. Coupled with the sensitivity of magnetoresistive (MR) sensors, the possibility of a lab-on-chip device, and the specificity of phage receptor binding proteins (RBPs) was constructed as probes for the quick diagnosis of *Enterococcus* and *Staphylococcus* with possible antimicrobial agent resistance [104]. The multicomponent nucleic acid enzyme-Au NP (MNAzyme-GNP) device was used in patients with central line related bloodstream infectious disease and revealed a sensitivity and specificity as high as 86% and 100%, respectively [105]. MRSA was detected in patient swabs with 90% clinical sensitivity and 95% clinical specificity [105]. Detecting *mecA* resistance genes in uncultured nasal, groin, axilla, and wound swabs from patients was done with 90% clinical sensitivity and 95% clinical specificity [105]. Rapid detection of bacteria with multiple drug resistance had been achieved by combining electrochemical immunoassays (EC-IA) for pathogen identification with confirmatory quantitative mass spectral immunoassays (MS-IA) based on signal ion emission reactive release amplification (SIERRA) nanoparticles with unique mass labels [88]. The nanotechnology-based detection method has been shown to provide a more efficient and more accurate way of detecting multi-drug resistant bacteria.

### 4.3. Advances in Nanotechnology in Treating Bacterial Infectious Diseases with Possible Antibiotic Resistance

Nanotechnology-based therapeutic methods have revealed an effective therapeutic result against multi-drug resistant bacterial infection. There are some examples of advances in nanotechnology in treating bacterial infectious diseases with possible antibiotic resistance.

#### 4.3.1. Silver (Ag) NPs

Ag NPs have shown synergic action with colistin, and imipenem against pandrug-resistant *A. baumannii* isolated from patients [106]. The efficacy of combined neomycin, an antibiotic with Ag-NPs in bacteria-infected mice with burn wounds revealed good healing ability with excellent wound contraction when the neomycin-Ag-NPs were used in the spray method [107]. Ag NPs@organic frameworks/graphene oxide in sericin/chitosan/polyvinyl alcohol hydrogel were constructed for the redox ability of tyrosine residues in silk sericin without extra chemicals [108]. The composite material was noted to be an ideal dressing for accelerating hemostasis, preventing multi-drug resistant bacterial infection and promoting wound healing [108]. Based on the good antibacterial ability of Ag NPs combating numerous bacterial strains compared with other antimicrobial agents, the IPM@AgNPs-PEG-NOTA nanocomposite (Ag NPs covered with SH-PEG-NOTA and loaded by imipenem) was developed [109]. The IPM@AgNPs-PEG-NOTA is a pH-sensitive nanodrug device with good efficacy in reducing resistance and is synergistically active against carbapenem-resistant *A. baumannii* [109].

#### 4.3.2. Nanotechnology Based Phage Therapy

Bacteriophages, or “phages” can attack bacteria, takeover their intracellular mechanism to let themselves survive, and thus kill the bacteria. Phage therapy has been proposed as good weapon in combating bacteria with multi-antimicrobial resistance [110]. However, how to prepare and deliver this “genetic material” into a human body is a major problem so far [110]. With the nanotechnology-based method, the phage might be delivered after remodeling with lipid based nano-carriers, microfluidic-based methods, nano-emulsification, or phage loaded nanofibers [111]. The advances could improve the stability of phage therapy, enhancing host retention times, and facilitate the biofilm penetration [111].

#### 4.3.3. Development of Nano-Cargos to Deliver Antimicrobial Agents

Surface engineering of nano-cargos has the advantages of aiming and controlling the antimicrobial-resistance mechanisms, and thus possibly overwhelms the bacterial resistance [112]. The incorporation of ivacaftor-colistin nanosuspensions (Iva-Col-NPs) was effective against the multi-drug resistance of *P. aeruginosa*, an overwhelming cause of terminal and persistent lung infections in cystic fibrosis patients [113]. Chitosan propolis nanocomposite coupled with apramycin was used for therapy for multiantimicrobial agent-resistant *S. Typhimurium* [114]. Nano-encapsulated daptomycin has been used in therapy for an experimental MRSA bone and joint infection model [115].

#### 4.3.4. Nanotechnology-Based New Antimicrobial Agent Delivery

Some new antimicrobial agents might be effective in combating antimicrobial agent resistant bacteria, but the clinical use might be limited because of the hydrophobicity or low oral bioavailability. A nanotechnology-based delivery method might provide a practicable way for therapy with these new antimicrobial agents. Platensimycin (PTM), a natural agent, could target bacterial fatty acid synthases and treat infections by MRSA [116]. To enable the usage of PTM in local MRSA infections, polyacrylamide hydrogels containing polyamidoamine was constructed for the organized release of PTM [116]. Luteolin-loaded methoxy poly-poly micelles could overcome the hydrophobicity and low oral bioavailability characters of luteolin. The luteolin-loaded NP has been shown to be effective in treating pulmonary infection with multi-antimicrobial agent resistant *K. pneumoniae* [117].

#### 4.3.5. Chitosan-Based Nanomaterial

Chitosan, itself an antimicrobial polysaccharide, is produced to be readily soluble in neutral aqueous solution during systemic therapy [118]. After modification, the acid-transforming chitosan (ATC) become an insoluble material in a slight acidic intracellular environment [118]. ATC, coupled with fragment DNA could form nano-sized spherical polyplexes and combat the *S. typhimurium* in macrophages [118]. Chitosan-Au loaded with embelin has shown synergistic activity with ciprofloxacin, an antimicrobial agent against multidrug-resistant *P. aeruginosa* and *E. coli*, by inhibiting the efflux pumps [119].

## 5. Clinical Applications of NPs for Bacterial Infections in Various Systems

Recent advances in the development of NPs to treat bacterial infectious diseases are summarized in Table 1. NPs have been used for the treatment of bacterial infections in clinical situations, with applications across different systems, as described below (Figure 3).

### 5.1. Central Nervous System: Transport of Antimicrobial Agents across the Blood–Brain Barrier (BBB)

When treating patients with central nervous system (CNS) infection, the most important challenge for drug delivery is the ability of antimicrobial agents to penetrate the BBB [138]. The BBB is highly selective and serves as the interface between the central nervous system and circulation, which is critical for maintaining brain homeostasis [138]. Since most of the pathogens infect the CNS through phagocytosis or receptor (e.g., EphA2)-mediated transcytosis, the majority of the NPs cross the BBB via receptor-mediated transcytosis (e.g., antibody, peptide, or protein) [139]. In neonatal meningitis caused by *E. coli* K1 infection, a chitosan-modified poly(lactic-coglycolic acid) (PLGA) nanoparticle was used as a vector for the recombinant protein OmpAVac (Vo), inducing protective immunity against *E. coli* K1 infection [120]. Transcranial picosecond laser stimulation of Au NPs after an intravenous injection has been noted to increase BBB permeability, which might be specifically graded by laser intensity and entirely reversible, without inducing significant structural damage [138]. For the treatment of CNS infections, drug-loaded NPs are safe and could be up to ten-fold more efficient than drugs alone [140]; therefore, nanotechnology is anticipated to be useful as vectors transporting antimicrobial agents across the BBB and subsequently provide new avenues for drug screening and therapeutic interventions in the central nervous system. However, the main bacterial pathogens infecting the CNS are *S. pneumoniae*, *H. influenzae*, and *N. meningitides* [141]. The efficacy of nanomedicine toward meningitis caused by these common pathogens remains unclear. Moreover, the most troublesome problem in the management of meningitis is the difficulty in detecting pathogens in the CNS. Methods to apply nanotechnology in the diagnosis of CNS infections should be analyzed in the future.

### 5.2. Respiratory System: Combatting Multidrug Resistant Pathogens at High Concentrations

Respiratory tract infections, especially pneumonia, are often caused by some multidrug-resistant bacterial strains. MDR may be overcome through the nanoparticle-mediated delivery of high-concentration antibacterial agents at the site of infection [16,122,123]. A dextran-coated stimuli-responsive nanoparticle that encapsulated the hydrophobic antibiotic rifampicin had a strong affinity for a variety of pathogens and effectively accumulated in bacteria-infected tissues. NPs were activated by either low pH or high concentrations of reactive oxygen species in the infectious microenvironment and released both cationic polymers and rifampicin that exerted synergistic effects on pneumonia in a mouse model of drug-resistant *P. aeruginosa* [16]. Antibacterial peptides loaded into porous silicon NPs (pSiNPs) exert a therapeutic effect on lung infection caused by *P. aeruginosa,* resulting in an obvious reduction in the bacterial load and improved survival in a mouse model [123].

Chemically engineered polyvinylpyrrolidone (PVP)-capped Ag NPs showed antibacterial activity against carbapenem-resistant strains of *A. baumannii*, a critical pathogen causing nosocomial respiratory tract infections. Ag NPs decreased the number of intracellular bacteria and bacterial adherence without any cytotoxic effects on human pulmonary cells [122].

Multidrug resistant, high mortality *Burkholderia mallei* is associated with weaponization by inducing aerosol inhalation, which results in glanders [98]. Au NPs functionalized with a glycoconjugate vaccine covalently conjugated to one of three different protein carriers (TetHc, Hcp1 and FliC) followed by conjugation to lipopolysaccharide (LPS) purified from a nonvirulent clonal relative, *B. thailandensis,* protect against lethal inhalations of *B. mallei* [121].

NPs have also been utilized for the accurate diagnosis of respiratory infectious disease [44]. LAMP-LFB provides simple, reliable, and objective detection of *M. pneumoniae*, one of the most common pathogens causing respiratory tract infection. This method has been particularly effective in the diagnosis of community-acquired pneumonia in school-age children [62].

Nanomaterials have been widely used to detect or treat bacterial infections in the respiratory system. In an era of respiratory tract infection with multidrug resistant pathogens, the high concentration of the drug delivered by NPs is expected to have more potent antibacterial activity. However, the main nanotechnologies used in the diagnosis or treatment of the respiratory system target only one or a few specific pathogens. In the real world, some of the pathogens in the respiratory system are multidrug resistant and polymicrobial, indicating that infection is caused by many bacteria. The treatment of multidrug-resistant and polymicrobial pathogens in the respiratory system will be the aim of future investigations.

### 5.3. Gastrointestinal System: Bypassing the Acidic Environment of the Stomach

The acidic environment of the stomach is an important defense mechanism of the human body, which eradicates ingested pathogens, but this acidity is also a threat to the therapeutic efficacy of antibiotics or probiotics [19,127]. The construction of delivery vectors that allow therapeutic agents to bypass the acidic environment of the stomach has always been a central aim in the development of NPs for treating bacterial infections in the gastrointestinal system [19,127]. *Helicobacter pylori* is the most well-known pathogen associated with gastric ulcers and gastric cancer, but the acidity of the stomach limits the therapeutic efficacy of antibiotics that eradicate *H. pylori* at the infection site [127]. Urea-modified UCCs-2 were used as a targeting moiety for UreI channel proteins that are specifically expressed on *H. pylori*. These pH-sensitive amoxicillin-loaded AMX-PLGA/UCCs-2 NPs protected antimicrobial drugs from the acidic environment of the stomach, safely delivering them to the *H. pylori* infection site [127].

Probiotics, such as *Bifidobacterium*, are vulnerable to gastric juices or gastrointestinal enzymes, and thus encapsulation is crucial to bypassing erosion and reaching target environments in the intestine or colon [19]. After encapsulation within alginate hydrogel-based shells, *B. breve* protected against simulated gastric acid and antimicrobial agents (tetracycline), thus successfully eradicating *E. coli* adhering to intestinal epithelial layers and maintaining surface coverage of Transwell membranes and their barrier integrity [19].

One of the most widespread toxin-mediated gastrointestinal pathogens is *V. cholerae*, which induces diarrhea by secreting cholera toxins [128]. The key host receptor for cholera toxin, monosialotetrahexosylganglioside (GM1), was coated onto the surface of polymeric NPs to function as toxin decoys. These NPs selectively and stably bound cholera toxins, neutralizing their actions on epithelial cells [128]. Another poly D,L-lactide-coglycolide (PLGA) nanoparticle encapsulated honeybee (*Apis mellifera*) venom and was capable of promoting the clearance of *S. enterica* serovar *Typhimurium* infection through the upregulation of T helper type 1-specific immune responses [125].

An antimicrobial peptide, VG16KRKP (VARGWKRKCPLFGKGG), delivered by gold nanoparticles (Au-VG16KRKP) was designed to tag against *S. typhi*, a common intestinal pathogen. These NPs were characterized as noncytotoxic to eukaryotic cells but exhibited strong bacteriolytic activity against intracellular *S. typhi* [126]. Another leading cause of diarrhea worldwide, *Shigella flexneri* (Shigellosis) was fortunately prevented by some outer membrane vesicles (OMVs) [124]. OMVs encapsulated in poly(anhydride) NPs were successfully developed for mucosal protection against *S. flexneri* when delivered through the nasal route [124].

NPs have been widely used as delivery vectors for therapeutic agents to bypass the acidic environment of the stomach as treatments for bacterial infections in the gastrointestinal system. The gastrointestinal system is a very complex environment composed of the microbiome and metabolome. The effects of NPs on the microbiota and its metabolites in the gut requires further evaluation.

### 5.4. Skeletal System: Effective in Bone Regeneration

Many NPs with bactericidal effects have been used to treat bone infection (such as osteomyelitis) or as part of the components of implants in the skeletal system [15,129,130,131]. Bioactive nanostructured CaSi layers on titanium substrates constructed by electrospray deposition were shown to exert antibacterial effects against gram-negative *E. coli* and gram-positive *S. aureus* species, with additional osteogenic properties when coated on bone tissue [15]. Calcium-alginate (Ca-Alg) NPs crosslinked with phosphorylated polyallylamine (PPAA) and prepared by salting-out achieved 82.55% encapsulation of clindamycin, an antimicrobial agent for bone *S. aureus* infections [131]. This Ca-Alg/PPAA/clindamycin nanoparticle was effective against MRSA osteomyelitis, with benefits in bone regeneration that prompted the recovery of infected fragments [131]. Another chitosan (CS)-based thermosensitive hydrogel was engineered to support vancomycin, an antimicrobial agent, in an osteomyelitis-NP/gel local drug delivery system. Vancomycin-NPs produced with quaternary ammonium chitosan and carboxylated chitosan NPs using positive and negative charge adsorption promoted osteoblast proliferation and were effective against osteomyelitis caused by *S. aureus* infection [129]. Silver-copper-boron (ACB) NPs were also used to treat infected osteoblasts by targeting intracellular *S. aureus* in bone cells, decreasing systemic toxicity and forming the basis of targeting specific markers expressed in bone infections [130]. In addition to antimicrobial effects, NPs are effective at promoting bone regeneration during the treatment of bone infection or as part of the components of implants in the skeletal system ex vivo and in vivo. The beneficial effect of its application in clinical patients remains unclear.

### 5.5. Skin and Soft Tissue: Targeting Bacteria and Their Biofilms

*S. aureus* is the most noticeable threat contributing to skin and soft tissue infection worldwide, with the capacity to form biofilms during infection [16,17,85]. Near-infrared-light (NIR)-activatable deoxyribonuclease (DNase)-carbon monoxide (CO) MPDA NPs successfully delivered bactericidal CO gas, penetrating impaired *S. aureus* biofilms and effectively eliminating them through the collaborative therapeutic effect of DNase I and CO gas-potentiated photothermal treatment [20]. A dextran-coated stimuli-responsive NP that encapsulated the hydrophobic antibiotic rifampicin accumulated in bacteria-infected tissues and was activated by either low pH or high levels of reactive oxygen species. When activated, the NPs released both cationic polymers and rifampicin, showing synergistic effects on drug-resistant *S. aureus* soft tissue (thigh) infection in a mouse model [16].

Biocompatible enzyme-responsive Ag NP assemblies (ANAs) are high-efficiency targeted antimicrobial treatments for MRSA and have been utilized as effective wound dressings to accelerate the healing of wounds with MRSA infections [17]. A biomimetic NP-based antivirulence (detained staphylococcal α-hemolysin (Hla)) vaccine that efficiently induced high anti-Hla titers exhibits protective immunity against MRSA skin infections and subsequently decreases the invasiveness of MRSA, preventing dissemination into other organs [132].

Group A *Streptococcus* (GAS), along with its infamous pore-forming streptolysin O (SLO) toxin, is a major pathogen causing severely invasive skin infections, including necrotizing fasciitis [133]. The pharmacological appearance of red blood cell (RBC)-derived biomimetic NPs (“nanosponges”) sequestered SLO and blocked the capacity of GAS to injure host cells, thereby preserving innate immune function while increasing bacterial clearance in a mouse model of GAS necrotizing skin infection [133].

*V. vulnificus* could cause quick and severe tissue damage, with the mortality rate as high as 50% [134,142]. The antimicrobial peptide (AMP) HPA3PHis loaded onto a Au NP-DNA aptamer (AuNP-Apt) coupled (AuNP-Apt-HPA3PHis) was effective against *V. vulnificus* infection by damaging membrane integrity, and decreasing intracellular *V. vulnificus* numbers up to 90% [134].

NPs have the advantages of targeting bacteria and their biofilms during the treatment of skin and soft tissue infections, especially tissues infected with *S. aureus* in vivo. However, evidence of the effect of NPs on eradicating bacteria hidden in biofilms or implants in clinical patients is still limited. More clinical studies are needed to confirm the effect of NPs on eradicating bacteria in biofilms or implants.

### 5.6. Ophthalmology System/Prolonged Drug Release with Less Uncomfortable Sensations

Several ocular drug delivery systems, such as hydrogels, microparticles, nanoemulsions, microemulsions, and liposomes, have been analyzed for drug release in the eye and sustain therapeutic levels for a prolonged period [143]. Nanomaterial-containing eye drops with the advantage of less uncomfortable sensations, fewer blinking reflexes, and less blurred visions are needed to better treat infectious ophthalmological diseases [135]. Moxifloxacin (MFX, an antibiotic) and dexamethasone (DEX, a type of steroid)-loaded nanostructured lipid carrier (Lipo-MFX/DEX) mixed with collagen/gelatin/alginate (CGA) biodegradable materials (CGA-Lipo-MFX/DEX) have been used as anti-inflammatory agents for treating ocular diseases, with properties that sustain drug release for at least 12 h, inhibit pathogen microorganism growth and improve corneal wound healing [135]. Zinc oxide (ZnO)/cyclized polyacrylonitrile (CPAN) composite-loaded contact lenses showed broad-spectrum antibacterial properties with the capacity to selectively block UVB and UVA, as well as blue light, under the premise of ensuring hydrophilicity and a certain transparency [144]. Artificial eyes containing Ag NPs significantly inhibited the growth of *S. pneumoniae*, *S. aureus*, *P. aeruginosa, and E. coli* [145]. In addition to bactericidal effects, various emerging nanomaterials, such as NPs, nanowires, hybrid nanostructures, and nanoscaffolds, have been tested in mouse models for ocular tissue engineering and regeneration [146]. Although NP-containing eye drops or ocular equipment provide prolonged drug release with less uncomfortable sensations, their long-term effects on the cornea and other eye structures require further evaluation.

### 5.7. Dental System/Overcoming the Complex Environments That Are Susceptible to Insults

The oral cavity and oropharynx are complex locations that face physical, chemical, and microbiological damage and have pathological and cancerous variations [147]. Alveolar bone loss is a common problem that affects dental implant placement and is influenced by fibrotic tissue ingrowth and bacterial infection [136]. Micro/nanoencapsulants with unique structures and properties are more favorable drug-release platforms for treating dental diseases than conventional treatment approaches [147]. Ag coating methods provide bioactive collagen barriers that help guide bone regeneration, and they are currently being coupled with antibacterial and anti-inflammatory properties [136]. An Ag-NP-coated collagen membrane showed outstanding antibacterial effects on *S. aureus* and *P. aeruginosa*, as well as anti-inflammatory effects achieved by reducing the expression and release of inflammatory cytokines, including IL-6 and TNF-α, and inducing the osteogenic differentiation of mesenchymal stem cells that guide bone regeneration [136]. Micro/nanomaterials of bioactive glasses (BGs) with biocompatibility and osteogenesis-promoting agents were analyzed using Zn-doped bioactive glass micro/nanospheres for dental pulp capping to combat bacterial infection and stimulate tissue renewal [148]. NPs can be used in the oral cavity and oropharynx to overcome complex environments that are susceptible to physical, chemical, and microbiological insults. However, the long-term efficacy and safety of these NPs in the complex environment of the oral cavity of clinical patients warrant further evaluation.

### 5.8. Reproductive System (Sexually Transmitted Disease)/Increasing the Retention Time and Decreasing Discomfort

The incidence of reproductive system infections (sexually transmitted disease), especially vaginal infections, has recently increased, especially among women of reproductive age, and requires an effective concentration of topical antimicrobial treatment for its direct therapeutic action, reduced drug doses and side effects, and self-insertion [149]. However, alterations in the physiological conditions of the vagina significantly alter the effectiveness of vaginally delivered drugs [149]. Consequently, nanomedicine has been applied with topical antimicrobial agents to increase the retention time in the vagina and decrease discomfort [149].

*Neisseria gonorrhoeae*, the causative agent of gonorrhea, is an important target for the treatment of sexually transmitted disease. MtrE, the outer membrane channel of the highly conserved gonococcal MtrCDE active efflux pump, is a surrogate for vaccine targets because of its significance in defending *N. gonorrhoeae* from host innate effectors [137]. Thus, ferritin NPs (ferritin nanocages) have been utilized as vectors for MtrE in gonorrhea vaccines [137].

Although NPs have not been used as often as treatments for the reproductive system thus far, the use of nanomedicine to increase the retention time and decrease discomfort is expected in the future. However, the effects of these NPs on the fertility of clinical patients should be evaluated.

### 5.9. Reticuloendothelial System

The reticuloendothelial system (RES) is also called the macrophage system because of the common property of phagocytosis, where the cells engulf and destroy bacteria, viruses, and other foreign substances, as well as ingest worn-out or abnormal cells in the body. Since intravenously administered NPs are recognized and quickly cleared from systemic circulation by the RES [150], the use of activated phagocytes for NPs delivering antimicrobial drugs to treat infectious diseases is an ideal strategy. This approach has limited use, however, when fighting local infections [151]. Gupta et al. (2004) revealed that the surface coating of superparamagnetic iron oxide NPs with hydrophilic-hydrophobic polymeric compounds (PEG) can be used to deliver antimicrobial agents [152]. The result was attained by reducing plasma protein adsorption to the NPs and eliminating RES uptake [152,153]. Consistently, NPs modified with PEGylated polymers could reduce the adherence of serum proteins, coupled with opsonins, and cleared by the RES [154]. However, some of these antimicrobials agents, although effective against some bacteria, might be poisonous to mammalian cells [155,156]. Polymers have nonetheless been used in NP preparations to increase the appropriate host response to cationic antimicrobial agents in a balanced manner [157]. Although NPs have been used in the RES to facilitate eradication of systemic bacterial infection ex vivo and in vivo, the efficacy and adverse side effects of NPs on the RES system of clinical patients are still uncertain.

## 6. Advances in Developing Nanotechnology for Bacterial Infectious Diseases

Many methods have been shown to increase the function of NPs in preventing bacterial infection or improving wound repair, including encapsulation methods and material design [19], the incorporation of activation systems in NPs [16,20], and the addition of immune-activating adjuvants [158]. The protection strength of encapsulating shells, however, varies among encapsulation methods and materials [19]. Alginate gelation (intermediate interaction due to hydrogen bonding) provided better protection of probiotic *B. breve* against simulated gastric fluid and antimicrobial agents (tetracycline) than protamine-assisted SiO_2_ NPs, yolk−shell packing (weak interactions across a void), and ZIF-8 mineralization (strong interaction due to coordinated covalent bonding) [19]. The selection of methods to increase the function of nanotechnology depends on the purpose of the diagnosis or treatment of the bacterial infectious disease.

The hydrophilicity and roughness of NPs modulate their ability to prevent biomaterial-associated infections [90]. Au NPs that are hydrophilic/rough (water contact angle of 30 degrees/surface roughness of 118 nm) provide better cellular interactions, facilitating macrophage-mediated repair of damage [90]. Within the same nanomaterial, the incorporation of photothermal or photodynamic properties [20] or an activation system triggered by low pH or high concentrations of reactive oxygen species both improve antibacterial effects [16]. NPs containing both DNase I and CO gas-potentiated photothermal therapy are activated after permeating the impaired biofilms with the help of NIR to combat *S. aureus* hiding inside biofilms [20]. A dextran-coated stimuli-responsive NP that encapsulated a hydrophobic antibiotic has strong affinity for a variety of pathogens and effectively accumulates in bacterial infected tissues. NPs are then activated by either low pH or high concentrations of reactive oxygen species in the infectious microenvironment to release the payload of antimicrobial agents [16]. The modification of the hydrophilicity and roughness of the NPs influences their effect on the infectious microenvironment.

Combinations of biomaterials encapsulated in NPs might provide additive therapeutic effects on bacterial infectious diseases [159]. Combining the nitric oxide donor S-nitroso-N-acetyl-penicillamine with strong antioxidant cerium oxide NPs (CNPs) induced a synergistic therapeutic interaction that inhibited the growth of *S. aureus*, *E. coli* and *Candida albicans* [159]. Liposomal NPs have also been investigated for drug delivery, with advantages of increasing the cellular uptake of drugs, as well as stabilizing the physicochemical properties and bioavailability of both water-soluble drugs and drugs with poor water solubility [135]. Antibiotics incorporated in liposomal NPs prolong the drug retention time in the local area and have been used to treat ophthalmological diseases [135]. An enzyme-responsive NP has been shown to be highly effective at targeting MRSA. NPs were prepared from enzyme-responsive branched copolymers (BPs) synthesized by living radical polymerizations and underwent stable/collapsed transitions upon exposure to serine protease-like B enzyme proteins (SplB) secreted by MRSA [17]. The strategy of combining biomaterials is anticipated to provide additive therapeutic effects.

The addition of immune-activating adjuvants represents another method for enhancing the effect of NPs [158]. Invariant NK T (iNKT) cells are innate glycolipid-specific T cells restricted to the nonpolymorphic Ag-presenting molecule CD1d, and an iNKT glycolipid agonist, α-galactosylceramide (αGC), was used as an adjuvant for a human protective epitope [158]. An NP that carries αGC plus antigenic polysaccharides from *S. pneumoniae* αGC-embedded NPs was used to trigger murine iNKT cells and B cells while avoiding iNKT anergy [158]. NPs containing αGC plus *S. pneumoniae* polysaccharides provoked vigorous IgM and IgG responses and defended mice against lethal *S. pneumoniae* infections [158]. Magnetic NP/alternating magnetic field (MNP/AMF) hyperthermia combined with conventional antibiotics was more effective against *S. aureus* biofilms, inducing increased uptake of antibiotics to the bacterial cells and promoting the bactericidal activity of macrophages against intracellular bacteria via the generation of reactive oxygen species (ROS) [160]. The incorporation of immune-activating adjuvants might further potentiate the effect of NPs.

Modification of the hydrophilicity and roughness of the NPs, combinations of biomaterials, or incorporation of immune-activating adjuvants are predicted to increase the therapeutic effect of NPs without increasing the adverse effects. However, the selection of the appropriate methods to increase the function of nanotechnology depends on the aims of the diagnosis or treatment of the bacterial infectious disease.

## 7. Conclusions

In an era of bacterial infections characterized by increasing antimicrobial resistance, medicine is running out of weapons to combat these pathogens. Although some new antimicrobial agents have been developed in recent years, the emergence of bacterial resistance against these new drugs was reported a few years later. The development of NPs against these troublesome bacteria might be an alternative therapeutic choice with more bactericidal effects and less cytotoxicity. In recent years, numerous advances in the application of organic, inorganic and hybrid NPs to combat bacterial infectious diseases have been reported. SNAPs have excellent antibacterial activity and low toxicity; expanded use of star-shaped polymers for both the diagnosis and treatment of infectious diseases is expected in the future. To date, most developed NPs have been applied superficially on the body, such as in wound infections, skin and soft tissue infections, dental infections, or ophthalmological infections. Thus, the use of NPs inside the body to combat internal organ infections still has some challenges to overcome, including methods to guide the particle to the target organ or area, confirm the biosafety of the particle when administered inside the body, protect the particle against the damage of gastric juices or intestinal enzymes (for orally administered NPs) and confirm that encapsulated agents had higher efficacy than unencapsulated agents. Fortunately, many methods to increase the function of NPs in preventing bacterial infection or reducing cytotoxicity have been developed, including changing encapsulation methods and materials, incorporating activation systems, or adding immune-activating adjuvants. The construction of star-shaped polymers, for example, has shown the potential for increased antibacterial effects and decreased toxicity to mammalian cells. The development of appropriate NPs specific for the target organ using an effective delivery method will be a challenge to overcome when developing NPs as treatments for bacterial infections in the future.

## Figures and Tables

**Figure 1 pharmaceutics-13-01913-f001:**
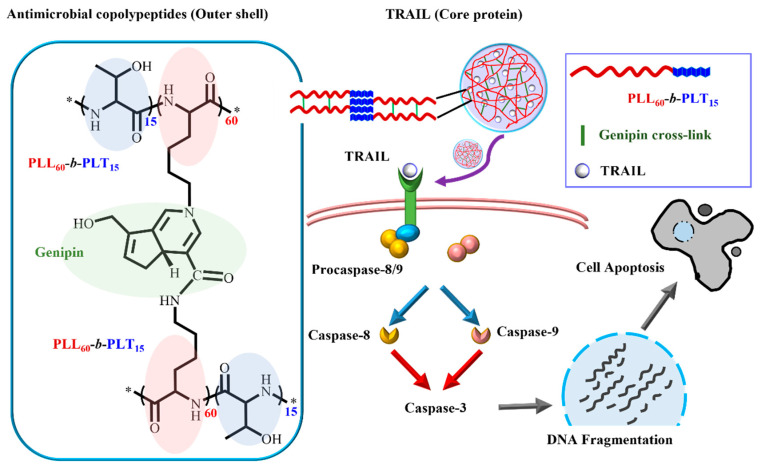
Schematic illustration of TRAIL encapsulated in bactericidal polypeptide nanogels, adapted from [36], published by Elsevier, 2019.

**Figure 2 pharmaceutics-13-01913-f002:**
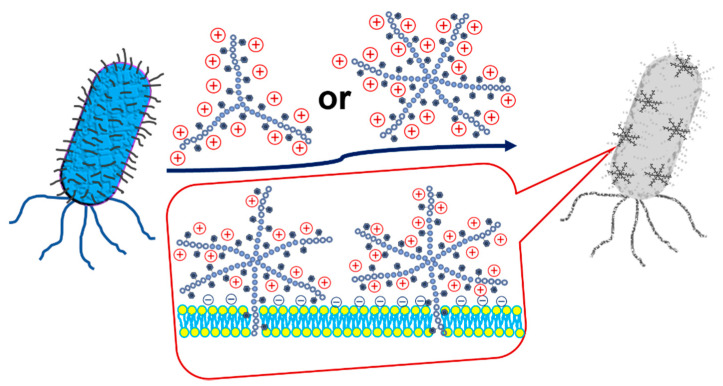
Schematic illustration of the antibacterial mechanism of star-shaped graft copolypeptides, adapted from [36], published by Elsevier, 2019.

**Figure 3 pharmaceutics-13-01913-f003:**
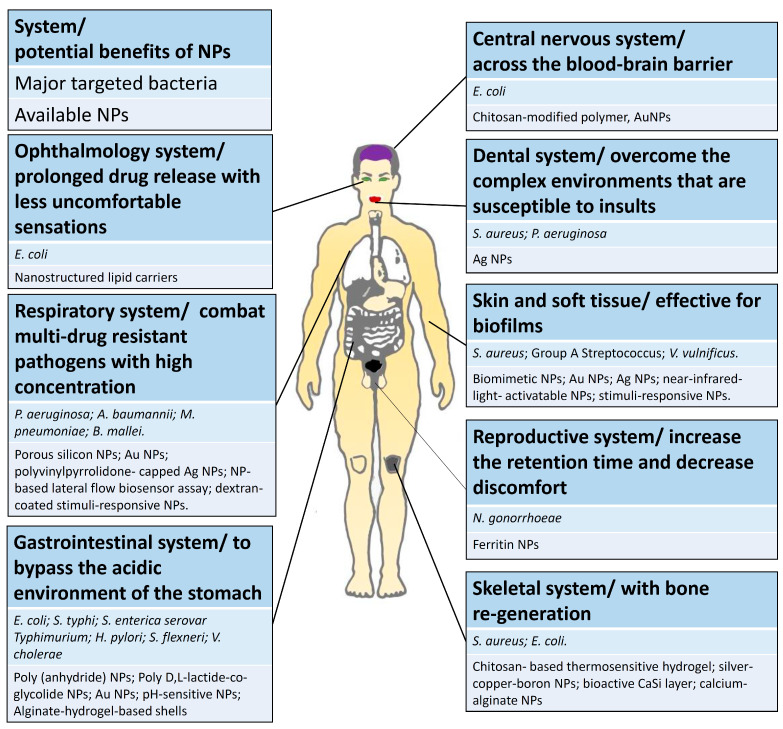
Advances in the use of NPs to treat bacterial infectious diseases in different body systems.

**Table 1 pharmaceutics-13-01913-t001:** Advances in the development of NPs to treat bacterial infectious diseases.

System/Disease	Targeted Pathogens	Material	Authors	Publish Year	Major Findings	References
Central nervous system						
Neonatal meningitis	*Escherichia coli* K1	Chitosan-modified poly (lactic-coglycolic acid) (PLGA)	Zhang J, et al.	2021	PLGA NPs as vector for the recombinant protein OmpAVac (Vo), which induced protective immunity against *E. coli* K1 infections.	[120]
Respiratory system						
Glanders	*Burkholderia mallei*	Gold NPs (AuNPs)	Gregory AE, et al.	2015	AuNPs functionalized with a glycoconjugate vaccines bind to LPS and protect against lethal inhalation of *B. mallei.*	[121]
Pneumonia	*Acinetobacter baumannii*	Polyvinylpyrrolidone (PVP)-capped Ag NPs	Tiwari V, et al.	2017	PVP-capped Ag NPs has shown antibacterial activity against a carbapenem-resistant strain of *A. baumannii*.	[122]
Pneumonia	*Pseudomonas aeruginosa*	Porous silicon NPs (pSiNPs)	Kwon EJ, et al.	2017	Selected antibacterial peptides loaded into pSiNPs exerted therapeutic effects on *P. aeruginosa* infections in the lung.	[123]
Pneumonia	*Mycoplasma pneumoniae*	Nanoparticle-based lateral flow biosensor (LFB) assay	Wang Y, et al.	2019	Loop-mediated isothermal amplification (LAMP) combined with LFBs for the rapid and accurate detection of *Mycoplasma pneumoniae*	[62]
Pneumonia	Drug-resistant *P. aeruginosa*	Dextran-coated stimuli-responsive nanoparticle	Ye M, et al.	2021	Encapsulation the hydrophobic antibiotic rifampicin, and strong affinity for *P. aeruginosa*, activated by either low pH or high reactive oxygen species levels.	[16]
Gastrointestinal system						
Intestinal infection	*Shigella flexneri* (Shigellosis)	Poly (anhydride) NPs	Camacho AI, et al.	2013	Outer membrane vesicles encapsulated poly NPs for mucosal protection against *S. flexneri.*	[124]
Intestinal infection	*Salmonella enterica* serovar *Typhimurium*	Poly D,L-lactide-coglycolide (PLGA) nanoparticle	Lee JA, et al.	2014	PLGA nanoparticle-encapsulated honeybee (*Apis mellifera*) venom promoted the clearance of *S. enterica* serovar *Typhimurium* infection.	[125]
Intestinal infection	*S. typhi*	Gold nanoparticle	Chowdhury R, et al.	2017	A designed antimicrobial peptide, VG16KRKP, delivered via gold nanoparticles exhibited strong bacteriolytic activity against intracellular *S. typhi*.	[126]
Stomach	*Helicobacter pylori*	Targeting the UreI channel protein, pH-sensitive NPs	Luo M, et al.	2018	Amoxicillin-loaded NPs protected antimicrobial drugs from an acidic environment, delivering them safely to eradicate *H. pylori* at the infected site.	[127]
Intestinal infection	*Vibrio cholerae*	Host receptor for cholera toxin was coated onto polymeric NPs	Das S, et al.	2018	The key host receptor for *V. cholerae* toxin, monosialotetrahexosylganglioside (GM1), was coated onto the surface of polymeric NPs and functioned as toxin decoys, neutralizing its actions.	[128]
Intestinal infection	*E. coli*	Alginate-hydrogel-based shells	Yuan L, et al.	2021	*Bifidobacterium breve*, when encapsulated within alginate-hydrogel-based shells yielded protection against gastric acid and antimicrobial agents, subsequently killing *E. coli* adhering to the intestinal epithelium.	[19]
Skeletal system						
Osteomyelitis	*Staphylococcus aureus*	Chitosan (CS)-based thermosensitive hydrogel	Tao J, et al.	2020	NPs containing vancomycin, an antimicrobial agent, combined with CS gel were effective against osteomyelitis caused by *S. aureus*.	[129]
Bone infection	*S. aureus*	Silver-copper-boron (ACB)	Abdulrehman T, et al.	2020	ACB NPs for bone infections that targeted intracellular *S. aureus* in bone cells.	[130]
Bone infection	*E. coli* and *S. aureus*	Bioactive CaSi layer on titanium substrate	Buga C, et al.	2021	A nanostructured CaSi layer exerts an antibacterial effect on *E. coli* and *S. aureus* species with osteogenic properties when coated on bone tissue	[15]
Osteomyelitis	Methicillin- resistant *S. aureus* (MRSA)	Calcium-alginate nanoparticle (Ca-Alg)	Gowri M, et al.	2021	Ca-Alg crosslinked phosphorylated polyallylamine (PPAA) encapsulating clindamycin, an antimicrobial agent, was effective against MRSA osteomyelitis.	[131]
Skin and soft tissue						
Skin infection	MRSA	Biomimetic nanoparticle	Wang F, et al.	2016	Nanoparticle-based detained staphylococcal α-hemolysin vaccine protected against MRSA skin infection and further decreased dissemination.	[132]
Necrotizing fasciitis	Group A *Streptococcus* (GAS)	Red blood cell derived biomimetic NPs (“nanosponges”)	Escajadillo T, et al.	2017	Nanosponges sequester pore-forming streptolysin O (SLO) and block the injury to host cells by GAS, preserving innate immune function and increasing bacterial clearance.	[133]
Wound infection	*Vibrio vulnificus*	Gold nanoparticle-DNA aptamer (AuNP-Apt)	Lee B, et al.	2017	An antimicrobial peptide (AMP), HPA3Phis, loaded onto AuNP-Apt reduced the intracellular *V. vulnificus* load by 90% and increased the viability of the infected cells.	[134]
Wound infection	MRSA	Biocompatible enzyme-responsive Ag nanoparticle assemblies (ANAs)	Zuo YM, et al.	2020	High-efficiency ANAs were an antimicrobial treatment for MRSA when applied as a wound dressing, accelerating healing.	[17]
Wound infection	*S. aureus*	Near-infrared-light (NIR)-activatable–carbon monoxide (CO)	Yuan Z, et al.	2021	Delivery of bactericidal CO gas that penetrated the impaired biofilms to achieve effective *S. aureus* biofilm elimination.	[20]
Soft tissue infection	*S. aureus*	The dextran-coated stimuli-responsive nanoparticle	Ye M, et al.	2021	Encapsulation of the hydrophobic antibiotic rifampicin, and then accumulating in infected soft tissues to eradicate drug-resistant *S. aureus* infection.	[16]
Ophthalmology						
Ocular wound	*E. coli*	Nanostructured lipid carriers mixed with a collagen/gelatin/alginate (CGA) biodegradable material	Chang MC, et al.	2020	Moxifloxacin (a type of antibiotic) and dexamethasone (a type of steroid)-loaded nanostructured lipid carriers for anti-inflammatory ocular disease treatment.	[135]
Dental tissues	*S. aureus* and *P. aeruginosa*	Silver-nanoparticle-coated collagen membrane	Chen P, et al.	2018	Antibacterial effects against *S. aureus* and *P. aeruginosa*, effective anti-inflammatory properties, and induction of the osteogenic differentiation of mesenchymal stem cells.	[136]
Reproductive system	*Neisseria* *gonorrhoeae*	Ferritin nanoparticle	Wang L, et al.	2017	Ferritin nanoparticles were utilized as a vector for MtrE, the outer membrane channel of gonococcal MtrCDE active efflux pump, in gonorrhea vaccines.	[137]

## Data Availability

Not applicable.

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
