# Peer review of "Advances in the Application of Nanomaterials as Treatments for Bacterial Infectious Diseases"

_pharmaceutics, 2021, doi:10.3390/pharmaceutics13111913_

Round 1
Reviewer 1 Report
The manuscript has been significantly improved.
Author Response
Dear the editor and reviewers of Pharmaceutics,
Enclosed, please find our revised manuscript entitled " Advances in the application of nanomaterials in bacterial infectious diseases ". We have considered very carefully for the concerns raised by the reviewers and made responsive alterations in the revised manuscript. We hope that our changes have satisfactorily clarified the points raised.
In answering the reviewers’ concerns, we made point-by-point responses to the comments summarized below. The revisions made in the manuscript were marked by red color, rather than track changes, because of the differences in the line numbers of the revised parts in the versions of “clear” manuscript and “track change” manuscript. We prefer to show the revisions in the clear manuscript in order to clearly indicate the revised parts by red color in the response letter. We appreciate the reviewers and the editor for their careful reading and insightful comments, and make our best efforts to improve the manuscript by responding to their comments.
We look forward to hearing from you.
Best Regards,
Jeng-Shiung Jan, jsjan@mail.ncku.edu.tw; Department of Chemical Engineering, National Cheng Kung University, No. 1, University Rd., Tainan 70101, Taiwan
Wen-Chien Ko, winston3415@gmail.com; Department of Internal Medicine, National Cheng Kung University Hospital, No. 138 Sheng Li Road, Tainan, 70403, Taiwan.
Reviewer 2 Report
After thorough corrections, I have no comments and I think that the manuscript is suitable for publication.Author Response
Dear the editor and reviewer of Pharmaceutics,
Enclosed, please find our revised manuscript entitled " Advances in the application of nanomaterials in bacterial infectious diseases ". We have considered very carefully for the concerns raised by the reviewers and made responsive alterations in the revised manuscript. We hope that our changes have satisfactorily clarified the points raised.
In answering the reviewers’ concerns, we made point-by-point responses to the comments summarized below. The revisions made in the manuscript were marked by red color, rather than track changes, because of the differences in the line numbers of the revised parts in the versions of “clear” manuscript and “track change” manuscript. We prefer to show the revisions in the clear manuscript in order to clearly indicate the revised parts by red color in the response letter. We appreciate the reviewers and the editor for their careful reading and insightful comments, and make our best efforts to improve the manuscript by responding to their comments.
We look forward to hearing from you.
Best Regards,
Jeng-Shiung Jan, jsjan@mail.ncku.edu.tw; Department of Chemical Engineering, National Cheng Kung University, No. 1, University Rd., Tainan 70101, Taiwan
Wen-Chien Ko, winston3415@gmail.com; Department of Internal Medicine, National Cheng Kung University Hospital, No. 138 Sheng Li Road, Tainan, 70403, Taiwan.

Reviewer 3 Report
Manuscript Number: Pharmaceutics-1425728
The review article manuscript of Yuan-Pin Hung, Yu-Fon Chen, Pei-Jane Tsai, I-Hsiu Huang, Wen-Chien Ko, Jeng-Shiung Jan “Advances in the application of nanomaterials in bacterial infectious diseases” presents the author opinion regarding the different types of nanomaterials and their current applications as treatments for bacterial infections characterized by possible antibiotic resistance and biofilms. Authors also emphasized the importance of nanomaterials in combating bacterial infections in different body systems.
The authors have presented an interesting review covering advances in the application of nanomaterials in the treatment of bacterial infectious diseases. Overall, this is an interesting study that addresses a key pharmaceutical problem today.
However, in reviewing the manuscript I have only few concerns. It seems that some of additional information has been latter added to the manuscript – a part of the text is in red colour. The following should be addressed when preparing a revision.
- At the beginning part of manuscript it could be necessary to add an information or separate additional paragraph about hybrid nanomaterials. If authors prepare a review article, it would be recommendable to provide more detailed information about hybrid nanomaterials. Authors discussed only one literature source about these nanomaterials– ref. 34 (lines 96-99), and the next one was mentioned only in lines 128-130.
- The same also regarding paragraph 3. Subparagraphs 3.1.1.-3.1.3. contained data only from 2-3 literature sources each, which is not enough for a review article. Please add additional information regarding mentioned topics. There is plenty of recent information regarding this topic.
- Authors underline in the introductory part that in review they also highlight the current applications as treatments for bacterial infections in regard to possible antibiotic resistance and biofilms. However the corresponding subtitles concerning possible antibiotic resistance and biofilms were not separately allocated in the manuscript. It would be recommendable to create separate paragraphs with related topics and provide information about it, as it was proposed in introduction.
Also many style and formatting errors that are needed to be reviewed.
Consequently, I do recommend accepting this manuscript for publication with major revision.
Author Response
Dear the editor and reviewer of Pharmaceutics,
Enclosed, please find our revised manuscript entitled " Advances in the application of nanomaterials in bacterial infectious diseases ". We have considered very carefully for the concerns raised by the reviewers and made responsive alterations in the revised manuscript. We hope that our changes have satisfactorily clarified the points raised.
In answering the reviewers’ concerns, we made point-by-point responses to the comments summarized below. The revisions made in the manuscript were marked by red color, rather than track changes, because of the differences in the line numbers of the revised parts in the versions of “clear” manuscript and “track change” manuscript. We prefer to show the revisions in the clear manuscript in order to clearly indicate the revised parts by red color in the response letter. We appreciate the reviewers and the editor for their careful reading and insightful comments, and make our best efforts to improve the manuscript by responding to their comments.
We look forward to hearing from you.
Best Regards,
Jeng-Shiung Jan, jsjan@mail.ncku.edu.tw; Department of Chemical Engineering, National Cheng Kung University, No. 1, University Rd., Tainan 70101, Taiwan
Wen-Chien Ko, winston3415@gmail.com; Department of Internal Medicine, National Cheng Kung University Hospital, No. 138 Sheng Li Road, Tainan, 70403, Taiwan.

Round 2
Reviewer 3 Report
Unfortunately after revision authors have not made any significant improvement of the manuscript and have not taken into the account the reviewer's recommendations. In the revised version, some paragraphs have been moved to another location, some have been marked in red color, without any change of text. Most of the colored texts are the same to the previous version.
Consequently, I do recommend accepting this manuscript for publication with major revision taking into the account my recomendations given in then the previous review.
Author Response
However, in reviewing the manuscript I have only few concerns. It seems that some of additional information has been latter added to the manuscript – a part of the text is in red colour. The following should be addressed when preparing a revision.
1. At the beginning part of manuscript it could be necessary to add an information or separate additional paragraph about hybrid nanomaterials. If authors prepare a review article, it would be recommendable to provide more detailed information about hybrid nanomaterials. Authors discussed only one literature source about these nanomaterials– ref. 34 (lines 96-99), and the next one was mentioned only in lines 128-130.
Reply: The authors thank the reviewer for the comment. The information about hybrid nanomaterials had been added at the beginning part of manuscript (line 47-52). We have reviewed 10 references related to the development of hybrid nanomaterials for treatments of infectious disease [4,5,37-44]. The detail description of examples of hybrid nanomaterials had been illustrated in lines 106-139 (highlighted in red) with the subtitles as shown in the following:
- Ag- based hybrid NPs
- Anti-bacterial agents-based hybrid
- Theranostics hybrid nanomaterial
- Bifunctional hybrid nano-flowers
2. The same also regarding paragraph 3. Subparagraphs 3.1.1.-3.1.3. contained data only from 2-3 literature sources each, which is not enough for a review article. Please add additional information regarding mentioned topics. There is plenty of recent information regarding this topic.
Reply: The authors thank the reviewer for the comment. We have added additional information regarding the mentioned topics lines 185-266 (highlighted in red) and rewritten this part to give a comprehensive review. As for the topic of biosensor, we discuss thoroughly with the subtitles as shown in the following:
3.1.1 Graphene-based biosensors
3.1.2. NP-based lateral flow biosensor
3.1.3. Sensitive label-free immunosensing
3.1.4. Nanosieving microfluidic system
3.1.5. Au NPs
3.1.6. Carbon nanotube biosensor
3. Authors underline in the introductory part that in review they also highlight the current applications as treatments for bacterial infections in regard to possible antibiotic resistance and biofilms. However the corresponding subtitles concerning possible antibiotic resistance and biofilms were not separately allocated in the manuscript. It would be recommendable to create separate paragraphs with related topics and provide information about it, as it was proposed in introduction.
Reply: The authors thank the reviewer for the suggestion. It has been separated as “4. Current applications of nanotechnology as treatments for bacterial infections in regard to possible antibiotic resistance and biofilms.”
As for medical device/efficacy against biofilms, it had been reviewed with the subtitles in lines 292-364 (highlighted in red) as shown in the following:
4.1.1 Application of nanotechnology in biofilms in biomaterial implants
4.1.2. Improve biofilm penetration with nanotechnology
4.1.3. Nanotechnology-base therapy in destroying biofilm formation
4.1.4. Biofilm Monitor with nanotechnology
4.1.5. Support the tissue regeneration in biofilms
As for “Advance in nanotechnology in treating bacterial infectious diseases with possible antibiotic resistance”, five important examples have been illustrated in lines 392-454 (highlighted in red) as shown in the following:
4.3.1 Silver (Ag) NPs
4.3.2 Nanotechnology based phage therapy
4.3.3. Development of nano-cargos to deliver antimicrobial agents
4.3.4. Nanotechnology- based new antimicrobial agent delivery
4.3.5. Chitosan-base nanomaterial
4. Also many style and formatting errors that are needed to be reviewed.
Reply: The style and formatting errors had been corrected.
5. Unfortunately, after revision authors have not made any significant improvement of the manuscript and have not taken into the account the reviewer's recommendations. In the revised version, some paragraphs have been moved to another location, some have been marked in red color, without any change of text. Most of the colored texts are the same to the previous version.
Consequently, I do recommend accepting this manuscript for publication with major revision taking into the account my recomendations given in then the previous review.
Reply: We appreciate the reviewers and the editor for their careful reading and insightful comments. We have made our best efforts to improve the manuscript by responding to their comments.

Round 3
Reviewer 3 Report
Accept in present form
This manuscript is a resubmission of an earlier submission. The following is a list of the peer review reports and author responses from that submission.
Round 1
Reviewer 1 Report
The manuscript presents different nanomaterials and various applications in bacterial infections. However, it is not clear what its originality is nor what is the story. The paper is badly written, confusing and scattered all around.
For instance, the title is “Advances in the application of nanoparticles in bacterial infectious diseases” but already on the second page, line 57 “nanogels” are mentioned, next, line 67 “nanostructured polymers”, then line “structuraly nano-engineered antimicrobial macromolecules”…
So what nanomaterial is the subject of this review? Definitely, nanoparticles are not, because one of the mostly used metal oxide nanoparticles are not considered.
Introduction is missing. Aim of the review is missing.
Another example is chapter “2.1. Diagnosis”. Only Au NPs are mentioned and only 2 tests. Taking into account that nanoparticles are widely used in biosensors, this chapter is not useless.
All chapters contain only a listing of published results without any comment.
Reviewer 2 Report
In the presented manuscript, the authors showed the use of various nanoparticles to combat bacterial infections. Basically, the approach taken in the study is interesting, and data may be appropriate for publication in Pharmaceutics after major corrections.
In my opinion, there are several threads missing from the manuscript.
In the manuscript, the authors omitted the problem of nanoparticle toxicity, which is a very important problem. In line 44, the authors even claim that AgNPs are relatively low-toxic.
Chapter 3.1. concerning the central nervous system, chapter 3.6. concerning ophthalmology system, chapter 3.7. concerning dental system and chapter 3.8. regarding reproductive system are too superficial and need to be expanded and more literature reported.
No explanation of the term STAR on line 77.